# Running after Activated Clotting Time Values in Patients Receiving Direct Oral Anticoagulants: A Potentially Dangerous Race. Results from a Prospective Study in Atrial Fibrillation Catheter Ablation Procedures

**DOI:** 10.3390/jcm10184240

**Published:** 2021-09-18

**Authors:** Karim Benali, Julien Verain, Nefissa Hammache, Charles Guenancia, Darren Hooks, Isabelle Magnin-Poull, Marie Toussaint-Hacquard, Christian de Chillou, Jean-Marc Sellal

**Affiliations:** 1Département de Cardiologie, CHRU de Nancy, 54500 Vandœuvre lès-Nancy, France; mirakbenali@gmail.com (K.B.); julienv13@free.fr (J.V.); h.nefissa@live.fr (N.H.); charles.guenancia@chu-dijon.fr (C.G.); i.magnin@chu-nancy.fr (I.M.-P.); c.dechillou@chru-nancy.fr (C.d.C.); 2INSERM-IADI, U1254, 54500 Vandœuvre lès-Nancy, France; 3Département de Cardiologie, CHU de Saint-Etienne, 42270 Saint-Priest-En-Jarez, France; 4Département de Cardiologie, CHU de Dijon, 21000 Dijon, France; 5Cardiology Department, Wellington Hospital, Wellington 6021, New Zealand; Darren.Hooks@ccdhb.org.nz; 6Laboratoire d’hémostase, CHRU de Nancy, 54500 Vandœuvre lès-Nancy, France; m.toussaint-hacquard@chru-nancy.fr

**Keywords:** anticoagulation, atrial fibrillation, heparin, activated clotting time, direct oral anticoagulants, vitamin K antagonist, anti-Xa activity, catheter ablation

## Abstract

Background: Activated Clotting Time (ACT) guided heparinization is the gold standard for titrating unfractionated heparin (UFH) administration during atrial fibrillation (AF) ablation procedures. The current ACT target (300 s) is based on studies in patients receiving a vitamin K antagonist (VKA). Several studies have shown that in patients receiving Direct Oral Anticoagulants (DOACs), the correlation between ACT values and UFH delivered dose is weak. Objective: To assess the relationship between ACT and real heparin anticoagulant effect measured by anti-Xa activity in patients receiving different anticoagulant treatments. Methods: Patients referred for AF catheter ablation in our centre were prospectively included depending on their anticoagulant type. Results: 113 patients were included, receiving rivaroxaban (*n* = 30), apixaban (*n* = 30), dabigatran (*n* = 30), and VKA (*n* = 23). To meet target ACT, a higher UFH dose was required in DOAC than VKA patients (14,077.8 IU vs. 9565.2 IU, *p* < 0.001), leading to a longer time to achieve target ACT (46.5 min vs. 27.3 min, *p* = 0.001). The correlation of ACT and anti-Xa activity was tighter in the VKA group (Spearman correlation ρ = 0.53), compared to the DOAC group (ρ = 0.19). Despite lower ACT values in the DOAC group, this group demonstrated a higher mean anti-Xa activity compared to the VKA group (1.56 ± 0.39 vs. 1.14 ± 0.36; *p* = 0.002). Conclusion: Use of a conventional ACT threshold at 300 s during AF ablation procedures leads to a significant increase in UFH administration in patients treated with DOACs. This increase corresponds more likely to an overdosing than a real increase in UFH requirement.

## 1. Introduction

Catheter ablation has become a widely used technique in the treatment of atrial fibrillation (AF) [1]. Thromboembolic events are among the most serious complications of the procedure [2]. These are mainly due to blood clot formation secondary to the presence of intracardiac catheters. To reduce this risk, optimal management of peri-procedural anticoagulation is fundamental. To avoid thrombi generation, oral anticoagulation is required at least one month before and two months after the procedure, using a vitamin K antagonist (VKA) or a direct oral anticoagulant (DOAC) [3]. Nevertheless, additional anticoagulation with unfractionated heparin (UFH) is necessary during the procedure to fully control this transient embolic risk. This combination of anticoagulants can expose patients to hemorrhagic complications, from femoral access bleeding to tamponade [4]. Since UFH dose required is highly variable between patients, Activated Clotting Time (ACT) guided heparinization has become the reference strategy for titrating anticoagulation during the procedure. Guidelines recommend maintaining ACT above 300 s to prevent embolic events [5].

Since 2009, DOACs have become an attractive alternative to VKAs, and are now used as first-line anticoagulants in AF [6]. The ACT target for patients treated with DOACs were directly extrapolated from data acquired in patients receiving VKA. However, there is considerable evidence to suggest strong differences between these two groups of patients. Numerous studies indicate that a significantly higher UFH dose is required to achieve an ACT > 300 s when patients received DOAC [7,8,9,10,11,12,13]. It remained unclear whether this difference was explained by a real higher requirement of UFH, or by an interference of DOACs with ACT reliability. In accordance with these considerations, we performed a prospective study to measure simultaneously ACT and heparin specific anti-Xa activity during AF ablation procedures in a population receiving either a VKA or a DOAC (factor II inhibitor and factor X inhibitor).

## 2. Materials and Methods

### 2.1. Study Design and Procedure Protocol

Patients referred for AF catheter ablation in our high-volume centre (>300 ablations/year) from October 2019 to October 2020 were prospectively enrolled. Patients were included depending on their oral anticoagulant, aiming to include 30 patients in each group: VKA, dabigatran, rivaroxaban, and apixaban. Oral anticoagulation was introduced at least one month before procedure. VKA were not interrupted. In the DOAC group, treatment was interrupted the day before the procedure and a continuous UFH infusion was prescribed to reach an activated partial thromboplastin time (aPPT) ratio between 2 and 3. At the beginning of the procedure, a 100 IU/kg UFH bolus was injected immediately after the transseptal puncture. ACT was monitored at baseline, 15 min after the initial UFH bolus, and additional UFH boluses were administrated to achieve an ACT target greater than 300 s. ACTs were measured 15 min after each additional bolus and at least every 30 min if no additional bolus was delivered. ACT was measured using a Hemochron Signature Elite^®^ (Accriva, San Diego, CA, USA) system. For each ACT, a blood sample was drawn simultaneously using the same venous femoral access. A citrate tube was collected and sent to the central laboratory: Anti-Xa activity was measured by a chromogenic anti-Xa assay (Stachrom^®^ Heparin, Stago, Asnieres-Sur-Seine, France). For rivaroxaban and apixaban, heparin specific anti-Xa activity was obtained by adding an activated charcoal, DOAC-stop^®^ (Haematex Research, Hornsby, Sydney, Australia), to overcome the interference of DOAC on coagulation [14]. Quantitative direct concentration of DOAC was also evaluated, using ecarin-based STA-ECA II^®^ (Stago, Asnieres-Sur-Seine, France) assay for dabigatran and Biophen DiXal^®^ (Hyphen Biomed, Neuville-Sur-Oise, France) for rivaroxaban and apixaban (Figure 1). The study was approved by local ethical committee (ref 220 CE-CHRUN) and all patients received oral and written information before the procedure.

### 2.2. Study Endpoints

We collected for each patient age, weight, type of oral anticoagulant, CHA2DS2-Vasc score, renal function, and concomitant drugs (including antiplatelet therapy). Bleeding and thrombo-embolic events during the procedure and the following 48 h were collected. We analyzed the correlation between ACT and anti-Xa activity for each sample, and whether this correlation was influenced by the type of oral anticoagulation.

### 2.3. Statistical Analysis

Continuous variables were expressed by mean and standard deviation (SD) and categorical variables by percentages. Comparisons between DOAC and VKA groups for clinical and procedural characteristics were performed by Fisher exact test and Chi-2 tests for categorical variables and Student-t and Wilcoxon tests for quantitative variables. Spearman rank correlation analyses (“ρ” coefficient) were performed to quantify the relation between pair samples of ACT and anti-Xa activity. The Fisher z-transformation was used to compare correlation coefficients between VKA and DOAC group. A *p*-value < 0.05 was considered statistically significant. Statistical analysis was computed using SAS 9.4 software (SAS Institute).

## 3. Results

### 3.1. Population and Procedural Characteristics

113 patients were included, receiving rivaroxaban (*n* = 30), apixaban (*n* = 30), dabigatran (*n* = 30) and VKA (*n* = 23). Among the 113 patients, most were male (71.7%), the mean age was 61.4 ± 10.6 yo and the median CHA2DS2-Vasc score was 1.7 ± 1.4. Mean left ventricular ejection fraction was 54.9 ± 9.0% and mean left atrial indexed volume was 41.9 ± 20.9 mL/m^2^ (42.0 ± 22.2 in the DOAC group, 41.3 ± 14.9 in the VKA group). The procedure was performed for paroxysmal AF in 52.2%, and 4.4% of the patients had concomitant antiplatelet therapy. Compared to the DOAC group, VKA patients were older (59.6 ± 10.8 vs. 68.4 ± 6.3), and had a higher CHA2DS2-Vasc score (1.5 ± 1.4 vs. 2.4 ± 1.3). The mean estimated glomerular filtration rate was 78.8 ± 13.0 mL/min/1.73 m^2^ (78.5 ± 13.6 in the DOAC group, 80.2 ± 10.4 in the VKA group). Baseline demographic and clinical characteristics of the population are illustrated in Table 1.

No bleeding or thromboembolic complications occurred in the two groups during the procedures or the following 48 h. 599 blood samples were collected, 482 in patients treated with DOAC, and 117 in patients treated with VKA. Mean INR in the VKA group was 2.5 ± 0.4, and mean DOAC concentration for rivaroxaban, apixaban, and dabigatran were 22.8 ± 30.2 ng/mL, 23.1 ± 16.9 ng/mL and 38.0 ± 19.8 ng/mL, respectively. There was no significant difference in the mean procedure duration between VKA group and DOAC group (165.4 ± 46.8 min vs. 176.5 ± 56.1 min, *p* = 0.38). Procedural data are described in Table 2.

### 3.2. UFH Dose, ACT, and Correlations with INR or DOAC Concentration

DOAC patients received significantly higher UFH doses than VKA patients (14,077 ± 4238 IU vs. 9565 ± 3629 IU, *p* < 0.001), but mean procedural ACT was less in DOAC patients compared to VKA (265 ± 44 vs. 294 ± 42, *p* = 0.007). The higher UFH dose in DOAC patients was reflected in a larger number of UFH boluses (3.3 ± 1.4 vs. 2.3 ± 1.1; *p* = 0.001), a longer time required to reach target ACT (46.5 ± 32.3 min vs. 27.3 ± 14.6 min, *p* < 0.001), and a higher proportion never reaching target (17% vs. 4%, *p* < 0.001). Despite the lower mean ACT in the DOAC group, this group demonstrated higher anti-Xa activity compared to the VKA group (1.56 ± 0.39 vs. 1.14 ± 0.36; *p* = 0.002) (Figure 2). Spearman correlation between UFH dose and anti-Xa activity was found to be close in the VKA group and in the DOAC group, with ρ = 0.73 (95%CI: 0.69 to 0.77) and ρ = 0.71 (95%CI: 0.67 to 0.75) respectively. The relationship between ACT and the intensity of oral anticoagulation differed according to the anticoagulant type. A good correlation was found between ACT values and INR for VKA patients (ρ = 0.57, 95%CI: 0.46 to 0.48) and with dabigatran concentrations for patients treated with factor IIa inhibitor (ρ = 0.48, 95%CI: 0.37 to 058). The correlation between ACT values and factor Xa inhibitor concentrations (apixaban and rivaroxaban) was found to be non-significant (ρ = 0.15, 95%CI: −0.04 to 0.30).

### 3.3. Correlation between ACT Values and Anti-Xa Activity

Spearman correlation between procedural ACT and anti-Xa activity in the VKA group was 0.53 (95%CI: 0.39 to 0.67). In the DOAC group (using DOAC-stop^®^ for factor X inhibitors), the Spearman correlation appeared weaker with ρ = 0.19 (95%CI: 0.09 to 0.30) (Figure 3). After Fisher z-transformation, the Spearman correlations between procedural ACT and anti-Xa activity for VKA and DOAC groups appeared significantly different (*p* = 0.006). In subgroup analysis, this correlation remained equally weak in patients treated with a direct factor Xa inhibitor (ρ = 0.17, 95%CI: 0.07 to 0.23), and patients treated with a direct factor IIa inhibitor (ρ = 0.22, 95%CI: 0.10 to 0.23) (Table 3).

## 4. Discussion

### 4.1. Main Findings

To the best of our knowledge, this study is the first to explore the correlation between ACT and anti-Xa activity in response to UFH administration during AF ablation procedures, according to anticoagulant type. From our results, the correlation between ACT and anti-Xa activity seems weak in the case of DOAC prescription (ρ = 0.19) while this correlation is acceptable when VKA are used (ρ = 0.53). This correlation remains weak in the two DOAC subgroups, both with factor Xa and factor IIa inhibitors. We also observed that total UFH dose delivered during a procedure was 46% higher in patients treated with DOAC compared with VKA patients. Paradoxically, the number of patients reaching the ACT target at 300 s was significantly smaller in DOAC patients, combined with a longer time to reach this target while anti-Xa activity was significantly higher in the DOAC group compared to VKA. All these results highlight the fact that ACT values seems to be misleading when a DOAC is present.

### 4.2. Correlation between ACT and the Intensity of Heparin Activity

The lower ACT threshold of 300 s recommended during AF ablation procedures was extrapolated from historical studies in patients treated by VKA [3,5]. This threshold was established to obtain the best compromise between hemorrhagic and thromboembolic risks [15]. Initially considered to be an equally appropriate threshold for DOAC, many studies recently established that patients treated with DOAC required more UFH to reach this ACT target at 300 s, a finding consistent with our results [7,8,9,13]. However, it remained unclear whether this difference was explained by a real higher requirement of UFH, or by an interference of DOAC with ACT measurement and reliability. In our current approach based on an ACT-guided heparinization strategy, the answer to this question is of considerable importance. Measurement of heparin activity by chromogenic anti-activated factor X (anti-Xa) assay is considered to be a reference method for monitoring of heparin therapy [16,17,18]. Our results show that the correlation between ACT and heparin activity represented by anti-Xa values seem to be correct in VKA patients, but appear to be very unreliable in DOAC patients (both with anti-II and anti-X drugs). It appears that this significant increase in UFH administration in patients treated with DOAC does not correspond to an increase in UFH requirement as demonstrated by the good correlation between UFH doses and anti-Xa activity in both groups, but clearly to a lack of reliability in ACT monitoring. More interestingly, the difficulty to reach the ACT target in patients receiving DOACs is associated with a significantly higher procedural anti-XA activity compared to patients receiving VKA. These contradictory results corroborate the hypothesis of a potential heparin overdosing during AF ablation procedures in patients treated with DOACs, already discussed by other authors [19]. Our study is also the first to explore anticoagulation parameters in a protocol with DOAC mini-interruption, a common practice in anticoagulant management during AF ablation procedures. Despite lower DOACs concentrations than in other studies [10,19], the relationship between ACT and anti-Xa activity remains weak in our population, showing that this interference is also present for low DOAC concentrations.

### 4.3. Clinical Implication

In our study, we did not observe any difference in the frequency of hemorrhagic or thromboembolic events between the DOAC and VKA groups. Most randomized controlled trials comparing DOAC and VKA safety during AF ablation did not observe any difference in the hemorrhagic complication rate, although a significant increase in UFH doses required in the DOAC group [7,8,20]. Given the constant improvement of the technical aspects and the very low rate of bleeding complications during AF ablation procedures, none of these studies were powerful enough to identify a potential clinical impact of this UFH overdosing. The ELIMINATE-AF trial, comparing the use of edoxaban to VKA during AF ablation procedures, demonstrated a trend towards a higher bleeding rate in the edoxaban group [9]. The authors emphasize that patients taking edoxaban received 24% more UFH and that most of the bleeding occurred during ablation or within the following 48 h, suggesting the role of this heparin overdosing and not of the oral anticoagulant. Edoxaban could not be analyzed in our study due to its non-availability in France.

### 4.4. Biological Considerations and Alternatives

ACT is a bedside functional assay used to monitor the inhibitory effect of heparin on secondary hemostasis. It measures the time required to activate the intrinsic and the common pathways of the coagulation cascade (Figure 4). ACT is expected to progress linearly with the dose of heparin delivered and is generally more accurate than aPPT to monitor high heparin doses [16,17,18,21,22]. Because of its availability, its low cost and its simplicity of application, ACT monitoring is actually widely used during interventional cardiology procedures or cardiovascular surgery to manage heparin administration [23,24,25]. However, there are several limitations in the routine use of ACT, notably the variability of reagents used on the different platforms (no gold standard system), the influence of patient characteristics on the test performance (platelet function abnormalities, antithrombin III level) or the use of concomitants drugs [26,27,28]. In our results, presence of DOAC clearly seems to interfere with ACT reliability. Recent in vitro results established that dose-response curves between UFH doses and ACT were parallel for patients treated with VKA or dabigatran, contrary to patients treated with rivaroxaban and apixaban [19]. In our study, although the correlation between ACT values at baseline and dabigatran concentrations appeared to be better than the correlation with rivaroxaban and apixaban concentrations, the correlation between procedural ACT values and anti-Xa activity in this group was also weak. This fact suggests the inability of ACT to reflect the real heparin therapy effect in patients treated with factor II inhibitors as well as patients treated with factor X inhibitors. The mechanisms underpinning the unreliability of ACT monitoring in the presence of DOACs are still unclear. Since monitoring of heparin therapy with ACTs appears to be appropriate for patients treated with VKA, but unreliable when DOACs are present, we should investigate alternative ways to adapt UFH administration. Efforts are needed to provide a new bedside coagulation assay compatible with DOACs [29]. In the meantime, we could consider alternative strategies, such as fixed protocols based on body weight or repeated direct anti-Xa measures during the procedure. This last strategy would appear to be the most interesting; however, the delay between the sample collection and results delivery to the clinician appears to be a major obstacle for routine use in short duration procedures such as AF catheter ablation.

### 4.5. Study Limitations

The present study has several limitations that must be considered. First, it is a single observational study. One could consider that some of the results depend on the laboratory assay used. Nevertheless, our data are concordant with the literature, indicating an increase in UFH dose used in patients treated with DOAC to reach a target ACT at 300 s. Due to the current higher DOAC prescription rate, we had greater difficulty in enrolling patients on VKA, with a smaller number of patients in this group. The mini-interrupted strategy commonly used in our centre could impact the influence of oral anticoagulation on ACT. However, consistently with previous studies, it appears that this correlation becomes weak as soon as a DOAC is present. Even though current guidelines recommend ablation with uninterrupted oral anticoagulation, many teams continue to hold off treatment the day before the procedure.

## 5. Conclusions

Use of a conventional ACT target at 300 s, resulting from a simple transposition of data acquired in the era of VKA, leads to a significant increase in UFH administration during AF catheter ablation procedures in patients treated with DOACs. This increase seems to correspond more likely to an overdosing than a real increase in UFH requirement. Our study shows in real life settings that the correlation between ACT values and anti-Xa activity appears unreliable as soon as a DOAC is present. Despite limited data on the clinical impact of this overdosing, clinicians should keep in mind this information during the management of intra-procedural heparin therapy. Further investigations are needed to clarify underlaying mechanisms, and to investigate new approaches to monitor heparin anticoagulant effect in patients receiving DOACs.

## Figures and Tables

**Figure 1 jcm-10-04240-f001:**
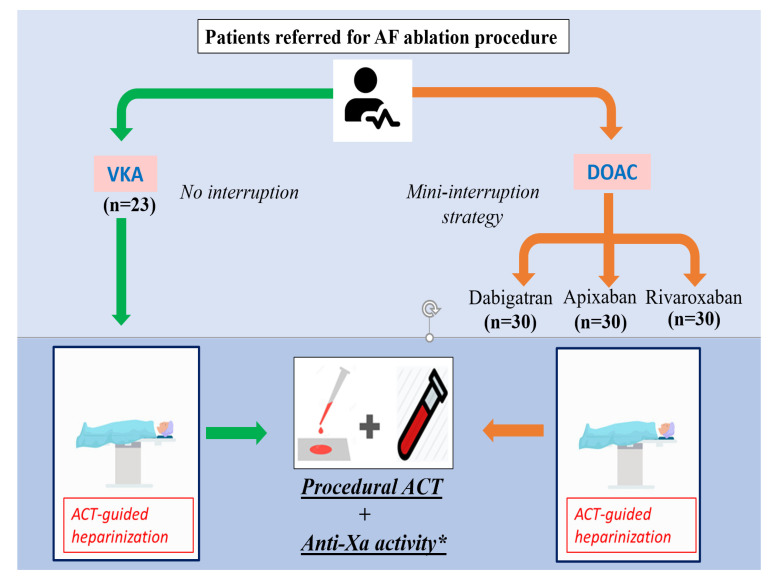
Study design. * *DOAC-stop* was used for Anti-Xa analyses for patients receiving factor X inhibitors.

**Figure 2 jcm-10-04240-f002:**
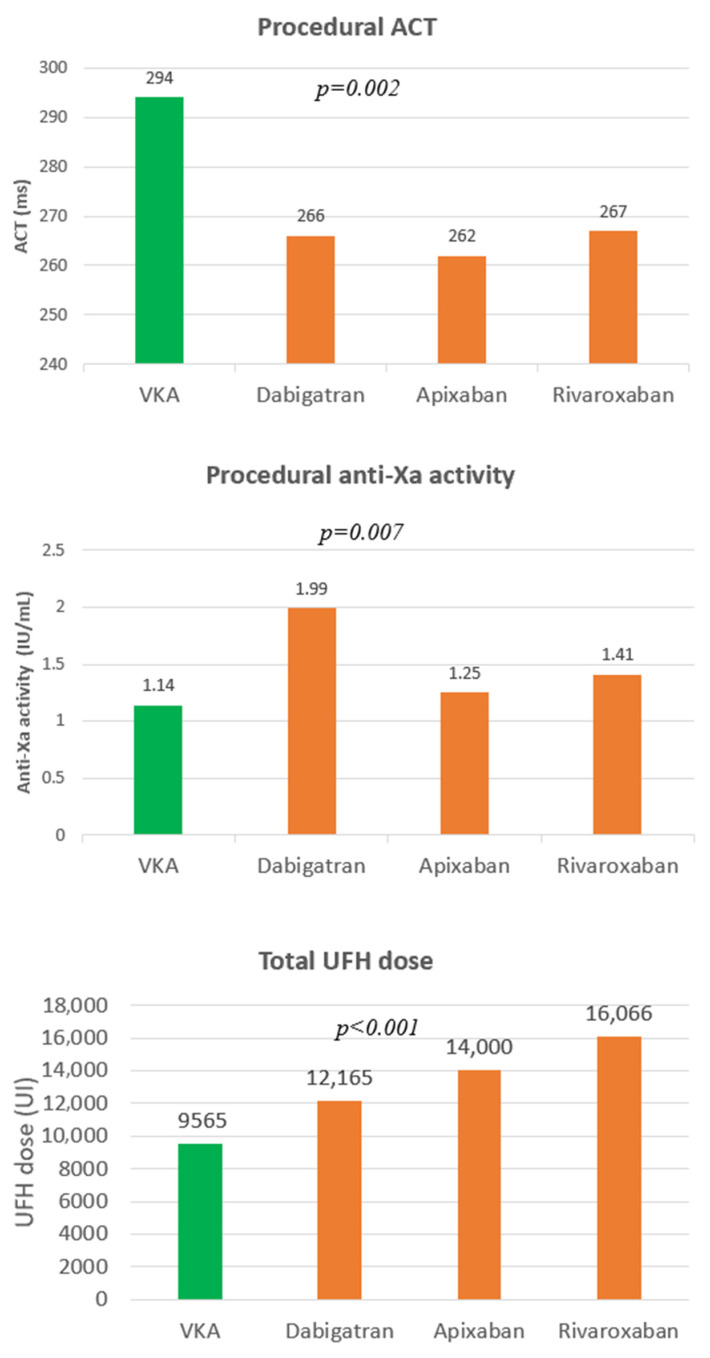
Mean ACT, anti-Xa activity, and UFH dose depending on anticoagulant type. The *p*-values represent those of the respective tests between VKA and DOACs groups.

**Figure 3 jcm-10-04240-f003:**
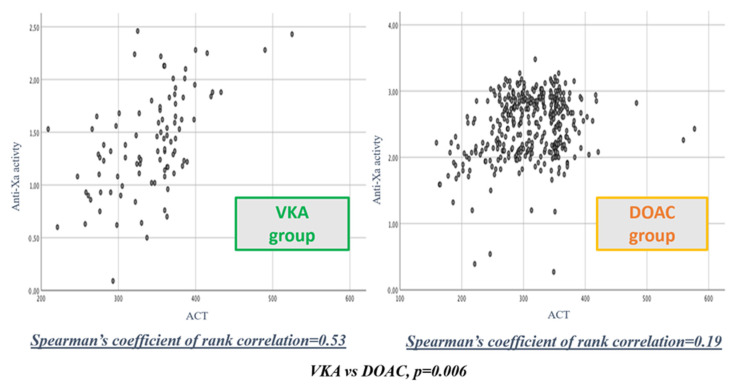
Spearman correlation coefficient between procedural ACT values and anti-Xa activity in patients receiving VKA or DOAC.

**Figure 4 jcm-10-04240-f004:**
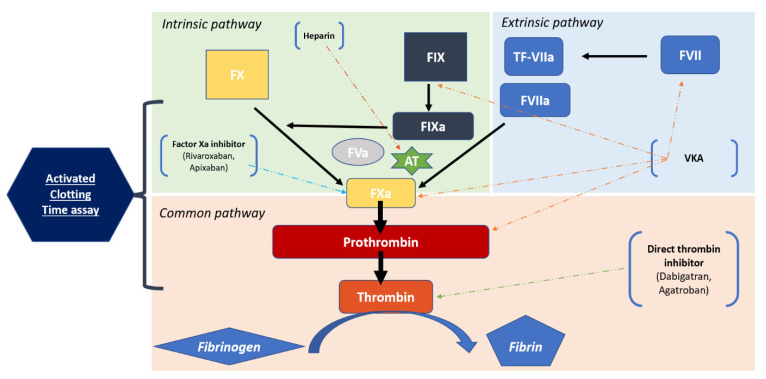
Coagulation process. AT = Anti-thrombin, TF = Tissue Factor.

**Table 1 jcm-10-04240-t001:** Population characteristics. Continuous variables are expressed by mean and standard deviation (SD) and categorical variables by percentages. LVEF = Left ventricle ejection fraction, Egfr = estimated glomerular filtration rate.

		Total		DOAC		VKA	
		*N* = 113		*N* = 90 (79.6%)		*N* = 23 (20.4%)	
		Mean ± SD	%	Mean ± SD	%	Mean ± SD	%
**Age, years**		61.4 ± 10.6		59.6 ± 10.8		68.4 ± 6.3	
**Male, sex**			71.7		74.4	60.9	60.9
**Weight, kg**		85.6 ± 17.6		86.8 ± 17.7		81.2 ± 17.0	
**Body mass index, kg/m^2^**		28.2 ± 4.8		28.3 ± 4.9		27.8 ± 4.7	
**LVEF, %**		54.9 ± 9.0		54.8 ± 9.7		55.4 ± 5.3	
**Left atrial indexed volume, mL/m^2^**		41.9 ± 20.9		42.0 ± 22.2		41.3 ± 14.9	
**CHA2DS2-VASc score, 0 to 9**		1.7 ± 1.4		1.5 ± 1.4		2.4 ± 1.3	
**Atrial fibrillation**							
	Paroxysmal		52.2		51.1		56.5
	Persistent		47.8		48.9		43.5
**Antiarrhythmic drugs**							
	None		7.1		8.9		0.0
	Amiodarone		30.1		27.8		39.1
	Flecainide		31.8		32.2		30.4
	Beta-blockers		65.5		62.2		78.2
	Verapamil		0.9		1.1		0.0
	Digoxin		1.8		1.1		4.3
**Antiplatelet therapy**							
	Aspirin or P2Y_12_ inhibitor		4.4		2.2		13.0
**Heart rhythm at procedure start**							
	Sinus Rhythm		59.3		56.7		69.6
**eGFR, mL/min/1.73 m^2^**		78.8 ± 13.0		78.5 ± 13.6		80.2 ± 10.4	

**Table 2 jcm-10-04240-t002:** Procedural characteristics. Continuous variables are expressed by mean and standard deviation (SD) and categorical variables by percentages. *: use of DOAC-stop^®^ in the blood sample.

	Total (*n* = 113)	VKA (*n* = 23)	DOAC (90)				*p*-Value (VKA vs. DOAC)
			Total DOAC (*n* = 90)	Rivaroxaban (*n* = 30)	Apixaban (*n* = 30)	Dabigatran (*n* = 30)	
**Baseline**							
-INR		2.5 (±0.4)					
-DOAC concentration (ng/mL)				22.8 (±30.2)	23.1 (±16.9)	38.0 (±19.8)	
-ACT (s)	157.5 (±33.4)	186.4 (±39.8)	150 (±27.1)	153.8 (±34.9)	143.7 (±22.0)	152.8 (±22.3)	<0.001
-Anti-Xa (IU/mL) (using DOAC-stop^®^ *)	0.52 (±0.29)	0.13 (±0.21)	0.62 (±0.22)	0.47 (±0.32) *	0.61 (±0.20) *	0.63(±0.17)	<0.001
**After 1st UFH bolus**							
-ACT (s)	271 (±45)	294 (±42)	265 (±44)	267 (±41)	262 (±36)	267 (±54)	0.007
-anti-Xa (IU/mL) (using DOAC-stop^®^ *)	1.88 (±0.54)	1.14 (±0.36)	1.56 (±0.39)	1.25 (±0.67) *	1.41 (±0.7) *	1.99 (±0.37)	0.002
-Total UFH dose (IU)	13,159.3 (±4493.2)	9565.2 (±3628.5)	14,077.8 (±4237.9)	16,066.7 (±4386.0)	14,000 (±1556.2)	12,166.7 (±2692)	<0.001
**ACT Targeting**							
-Nb UFH bolus to reach ACT > 300 s	3.1 (±1.4)	2.3 (±1.1)	3.3 (±1.4)	3.8 (±1.4)	3.2 (±1.4)	2.8 (±1.2)	0.001
-Time to ACT > 300 s, s (min)	42.3 (±30.3)	27.3 (±14.6)	46.5 (±32.3)	43.3 (±19.6)	47.5 (±20.3)	46.9 (±33.8)	<0.001
-Patients with ACT > 300 s reached	99 (87.6%)	22 (95.7%)	77 (85.6%)	25 (83.3%)	25 (83.3%)	26 (86.6%)	<0.001
-Procedure length (min)	174.2 (±54.3)	165.4 (±46.8)	176.5 (±56.1)	186 (±61.0)	170 (±51.8)	173 (±55.8)	0.38
**Peri-procedural Complications**							
Bleeding complications	0 (0%)	0 (0%)	0 (0%)	0 (0%)	0 (0%)	0 (0%)	1
Thromboembolic complications	0 (0%)	0 (0%)	0 (0%)	0 (0%)	0 (0%)	0 (0%)	1

**Table 3 jcm-10-04240-t003:** Correlation between UFH dose, procedural ACT values, and anti-Xa activity depending on the anticoagulant type.

	Spearman Correlation Coefficient	95%CI	*p*-Value
**Correlation between INR/DOAC concentration and ACT values at baseline**			
VKA group	0.57	0.46 to 0.68	0.004
DOAC group	0.32	0.15 to 0.46	0.021
-Factor Xa inhibitors (apixaban, rivaroxaban)	0.15	−0.04 to 030	0.055
-Factor IIa inhibitors (dabigatran)	0.48	0.37 to 058	0.007
**Correlation between UFH dose and anti-Xa activity**			
VKA group	0.73	0.69 to 0.77	<0.001
DOAC group	0.71	0.67 to 0.75	<0.001
**Correlation between procedural ACT and anti-Xa activity**			
VKA group	0.53	0.39 to 0.67	<0.001
DOAC group	0.19	0.09 to 0.30	0.022
-Factor Xa inhibitors (apixaban, rivaroxaban)	0.17	0.07 to 0.23	0.018
-Factor IIa inhibitors (dabigatran)	0.22	0.10 to 0.23	0.020
VKA vs. DOAC			0.006

## Data Availability

The data presented in this study are available on request from the corresponding author.

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
