# Peer review of "Running after Activated Clotting Time Values in Patients Receiving Direct Oral Anticoagulants: A Potentially Dangerous Race. Results from a Prospective Study in Atrial Fibrillation Catheter Ablation Procedures"

_jcm, 2021, doi:10.3390/jcm10184240_

Round 1
Reviewer 1 Report
In this original prospective study, the authors evaluate the differences in perioperative heparin anticoagulation during AF ablation procedure between patients on various anticoagulation patterns (VKA vs NOAC).
What is the main advantage of the paper is the fact that there is strictly limited data on the comparison of ACT and anti-Xa activity values subsequently to administering the standard dosage of heparin during the invasive procedures. The issue of proper heparin dosing during invasive procedures is of utmost importance. The authors draw attention to the possible necessity to change the anticoagulation measurement from ACT to another method depending on the prior received treatment. Unfortunately, it is a single centre study with a limited population, which requires further research to prove the outcomes and implement them in everyday clinical practice.
The manuscript is interesting and well written, although I have a few minor comments.
- Please consider improving the clarity of figure 1 – the print is too small in my opinion
- Could the authors please explain the difference between collected blood samples in pts on DOACs vs those on VKA (482 vs 117)?
Reviewer 2 Report
Benali et al. sought to analyze the differences in measurements of activated Clotting Time (ACT) values in patients receiving Direct Oral Anticoagulants (DOACs) compared to those who were on vitamin K antagonists (VKA). A prospective study including patients who were scheduled for atrial fibrillation catheter ablation procedures was planned and 113 patients were enrolled.
The authors found that in order to meet target ACT, a higher UFH dose was required in DOAC than VKA. The correlation of ACT and anti-Xa activity was tighter in the VKA group. Lower ACT values in the DOAC group were found and this group demonstrated a higher mean anti-Xa activity compared to the VKA group. They concluded that use of a conventional ACT threshold at 300s during ablation procedures can lead to a significant increase in heparin administration in patients treated with DOACs. This increase corresponds more likely to an overdosing than a real increase in UFH requirement.
The authors present a highly interesting study and I think the clinical relevance is high and the presented data is of great interest do readers.
Minor points:
I would recommend adding more demographic details, e.g. data on liver function parameters.
